# Enhancing Photocatalytic Properties of TiO_2_ Photocatalyst and Heterojunctions: A Comprehensive Review of the Impact of Biphasic Systems in Aerogels and Xerogels Synthesis, Methods, and Mechanisms for Environmental Applications

**DOI:** 10.3390/gels9120976

**Published:** 2023-12-13

**Authors:** Lizeth Katherine Tinoco Navarro, Cihlar Jaroslav

**Affiliations:** 1CEITEC-Central European Institute of Technology, Brno University of Technology, Purkynova 656/123, 612 00 Brno, Czech Republic; jaroslav.cihlar@ceitec.vutbr.cz; 2Institute of Materials Science and Engineering, Brno University of Technology, Technicka 2, 616 69 Brno, Czech Republic

**Keywords:** TiO_2_, anatase, brookite, sol-gel synthesis, heterojunctions, photocatalysis

## Abstract

This review provides a detailed exploration of titanium dioxide (TiO_2_) photocatalysts, emphasizing structural phases, heterophase junctions, and their impact on efficiency. Key points include diverse synthesis methods, with a focus on the sol-gel route and variants like low-temperature hydrothermal synthesis (LTHT). The review delves into the influence of acid-base donors on gelation, dissects crucial drying techniques for TiO_2_ aerogel or xerogel catalysts, and meticulously examines mechanisms underlying photocatalytic activity. It highlights the role of physicochemical properties in charge diffusion, carrier recombination, and the impact of scavengers in photo-oxidation/reduction. Additionally, TiO_2_ doping techniques and heterostructures and their potential for enhancing efficiency are briefly discussed, all within the context of environmental applications.

## 1. Introduction

TiO_2_, as a nanomaterial for aerogels and xerogels fabrication, is much studied due to its specific properties and essential role in lightening the environmental and energy crise through the effective use of solar energy and is presented as one of the most promising photocatalysts, mainly because of its significant physical and chemical properties such as high redox-potential, strong resistance to chemical and photo-corrosion, low cost, nontoxicity, stability, facile preparation with diverse morphologies, and significantly low energy consumption [1,2,3,4].

Nevertheless, a single-phase TiO_2_ as a catalyst still presents some limitations regarding the accumulation of electrons in the conductive band, leading to the recombination of the photoexcited electron hole [5]. The limitations of its application in the visible light spectrum represent a great challenge for and perspective on investment in water quality technologies in visible light, including decomposition of organic compounds and hydrogen production [6,7].

Research to date has provided advances in better photocatalytic properties through applying a biphasic system of TiO_2_; in contrast, with an SP anatase, brookite and rutile are used, with significant progress based on the modification of their crystal structure and particle size [7,8,9,10,11,12].

Since Fujishima and Honda’s (1972) revolutionary work, hydrogen research has been directed to water splitting on semiconductor photocatalysts [11]. The heterogeneous photocatalytic process is considered an excellent alternative to H_2_ production [13,14].

The biphasic system enhances water splitting by elevating oxidation levels. This intricate process involves complex reactions, employing organic compounds like methanol as sacrificial agents to overcome limitations. Methanol, serving as an electron donor, can be oxidized by photogenerated holes in the valence band, substituting for water [15]. Alternatively, carboxylic acid has proven effective as a sacrificial agent, successfully evolving hydrogen in certain photocatalytic reactions [16]. Photogenerated hydrogen is influenced by surface defects, including grain boundaries and interfacial interfaces, which serve as active centres for photocatalysis [17]. Additionally, the beneficial charge transfer between biphasic systems, such as anatase-rutile [18] and anatase-brookite [19], is noteworthy.

Frequent attempts to synthesize brookite and the other crystalline forms of titania using various techniques have been widely described by many authors in the literature [9,11,12,20,21,22,23]. The sol-gel technique is adequate due to its simplicity and cost-efficiency. Additionally, it allows improved crystallinity in comparison with other methods (high-energy synthesis methods such as sputtering and pulsed laser deposition PLD, [21]) and further enhances TiO_2_ characteristics for photocatalytic applications [11,24].

Other studies evaluated several precursors’ effects over crystal phase transformation in TiO_2_ systems during sol-gel synthesis. In recent years, the most often used precursors were TiCl_4_ or TiCl_3_ for synthesized pure brookite nanoparticles [8,25], nanotubes [4,15,18,26,27], nanorods [28,29], nanospheres [24] nanoplates [30], aerogels and xerogels [31]. Adjusting pH with aqueous HCl, NaOH, ammonia, and organic compounds from carboxylic acid or amine donors is also applied in other approaches to evolve crystal phases [32]. In 2015, Leyva-Porras et al. synthesized anatase TiO_2_ nanoparticles through an acid-assisted sol-gel method at 25 and 80 °C, specifically acetic acid. Moreover, positive effects were observed regarding particle size and brookite formation with the acid’s increment over the xerogel. The authors found a narrow particle size distribution with an increment for the brookite amount [20].

Thus, it is still under question how the pure biphasic system TiO_2_ in aerogels and xerogels is formed in acidic conditions influenced by organic substances. Since preliminary results show the effects of different chelate forming substances as a source of carboxylic acid donors, the presence of lactic acid drifts to the highest brookite phase composition in comparison with the other sol-gel complex synthesis [8]. A chelating donor agent is considered an essential factor influencing the biphasic system’s crystallography, subsequently prompting the material’s photocatalytic activity [33].

Sol-gel synthesis under atmosphere conditions is studied to explain the influence of the chelating agents as donors from organic substances on the formation and composition of TiO_2_ crystalline phases. This approach will also establish the organic dependence and selectivity during poetization on the growth of TiO_2_ on the phase transition and the photocatalytic efficiency of the biphasic systems.

Recent advancements in overcoming the limitations of titanium dioxide (TiO_2_) as a photocatalyst involve innovative strategies aimed at enhancing its efficiency. Doping TiO_2_ with non-metals (e.g., nitrogen) [34] or metals (e.g., silver, copper) remains a promising avenue, modifying its electronic structure and improving charge separation for heightened photocatalytic activity [35]. Surface modifications, such as coating TiO_2_ with materials like graphene or carbon-based substances, have been explored to enhance surface area and inhibit charge carrier recombination, leading to improved photocatalytic performance. Hybridizing TiO_2_ with other semiconductor materials, like ZnO [36] or CdS [24], to form heterojunctions, has been shown to enhance charge separation and overall photocatalytic efficiency. Strategies to extend TiO_2′_s absorption into the visible range, such as bandgap engineering or coupling with narrow-bandgap semiconductors, are being pursued to enhance its photocatalytic activity under solar light [37]. Introducing co-catalysts like Pt or Pd onto TiO_2_ surfaces improves charge transfer and enhances the efficiency of specific reactions [19,38]. Additionally, developing heterogeneous photocatalysis systems, including immobilizing TiO_2_ on various supports or incorporating it into membranes, aims to improve catalyst recovery, reusability, and stability [39,40]. These approaches collectively represent the forefront of efforts to address the limitations of TiO_2_, paving the way for its broader and more effective application in photocatalysis.

The review article aims to explore titanium dioxide (TiO_2_) photocatalysts, emphasizing the impact of their structural characteristics and heterojunctions’ on photocatalytic efficiency. It also aims to delve into various synthesis methods, such as aerogel and xerogel, mechanisms of photocatalytic activity, and the potential applications of TiO_2_ photocatalysts in environmental contexts. Ultimately, the review is valuable for researchers and practitioners seeking to understand and harness TiO_2_ photocatalysts for environmental improvements.

## 2. TiO_2_ as Photocatalyst and Heterojunctions

### 2.1. TiO_2_ Photocatalyst Advantages That Benefit the World Ecology and Everyday Health

Among many semiconductor photocatalysts, researchers have categorized TiO_2_ as a superior photoactive material due to several benefits during its application, such as high activity, excellent stability under light illumination, low price, and nontoxicity, which provide greater photocatalytic efficiency than other catalysts, such as α -Fe2O_3_, ZrO_2_, CdS, WO_3_ and SnO_2_ due to the necessity of using an external electrical bias to complete the water-splitting reaction [41,42]. The photocatalyst has high surface area efficiency and is environmentally friendly for hydrogen production using renewable resources such as solar energy to enable future energy supplies. Additionally, several authors summarized H_2_ production trough water splitting [16,43,44]. However, attempts to solve the absorption of visible light brought with them numerous approaches, including dye sensitization, doping, coupling, and coating of TiO_2_.

The photocatalyst’s disadvantages are expensive precursors, extended times of reaction, and carbon as an impurity during synthesis. Manipulation of TiO_2_ is easy its toxicity is low. However, various studies consider that material’s safety depends on the size and crystal form, which strongly determine the toxicity potential. The dangerous effects of TiO_2_ are associated with the called “overload” from inhalation exposure, which is unusual in everyday life. It appears that its toxicity can be modified by combining it with photosensitizers. Applying the photochemical activity of TiO_2_ improves tooth personal care and teeth bleaching. Additionally, its scaffolds are successfully used in the preparation of implants for surgery in bone tissue engineering. TiO_2_ works as a carrier in the pharmaceutical sciences, and its catalytic system can be used to eliminate dangerous chemicals and pollutants [41](Figure 1). The decomposition of organic compounds like methylene blue, methyl orange, toluene, p-nitrophenol, crocein orange G, methyl red, congo red, and stearic acid by using UV/Vis irradiation is frequent [45], as is degradation of pesticides such as malathion, and atrazine [46,47].

### 2.2. Structure of TiO_2_ Phases: Anatase, Brookite, Rutile

Titanium dioxide has three different crystal forms that naturally appear in nature [46]. Anatase and rutile are the two most common tetragonal crystallographic polymorphs. Brookite is rarer, with an orthorhombic crystal structure (Figure 2). The arrangement in each phase is represented by an octahedron constituting oxygen ions at its vertices (O^2−^) and titanium atoms at the centre (Ti^4+^) [49] with different spatial arrangements sharing the edges and corners in a different mode. There is a structural inter-relationship between anatase and rutile, where rutile is the most thermodynamically stable phase at all temperatures and pressures, while meta-stable anatase and brookite are kinetic products, as evidenced by theoretical and experimental studies [50,51,52].

#### Corner-Sharing and Edge-Sharing Modes of Arrangement

In the evolution of the titania polymorphs, the TiO_6_ octahedral in tetragonal anatase is surrounded by four shared edges and four shared corners [1,55], consisting of a body-centred to form zigzag ribbons along the lattice planes (2 2 1), while brookite adopts an orthorhombic structure, in which the octahedra share three edges with one of the edges along the (1 0 0) direction and the other two edges along the (0 0 1) lattice planes with tunnels along the c-axis [12]. At the same time, rutile is composed of corner-sharing octahedra, with each octahedron surrounded by ten octahedra, in which two are edge-shared, and eight are corner-shared its lattice shares two edges to form linear chains parallel to (0 0 1) [54].

### 2.3. Heterophase Junctions of Anatase-Brookite (Rutile) and Photocatalyst Efficiency

Depending on the application, some structural features of a polymorph will be more vital than others: size or surface area to volume ratio plays a central role in catalysis. Controlling their size and exposed facets is essential for fabricating photocatalytic crystals, providing more flexibility and options for designing nanostructures to satisfy unique requirements. The efficiency of a photocatalyst can be tuned by engineering the crystallinity, surface area, morphology, and bandgap of the materials [56,57,58].

A heterophase structure of titania is beneficial for green energy technologies due to the optimisation of the phase composition. The morphology at the polymorphic interfacing and surface states are acute for such a synergistic impact, and hint contact between the stages shows excellent effects on the charge carrier exchange in light-induced photoreactions. The different band gaps and the position of CB and VB edges result in the arrangement of steady heterophase junctions that permit the retention of a more extensive spectral range and move forward the charge carrier dynamic, and the separation and exchange process [58].

In this context, the presence of brookite helps to retard the recombination of holes and accumulation of electrons in the conductive band, leading to increasing the oxidation reactions. So, structuring a cooperative behaviour between the anatase and brookite phases is the essential characteristic of the TiO_2_ catalyst [21]. Brookite nanoparticles exhibit higher photo activity toward methanol photo-oxidation than anatase nanoparticles. This difference is explained by considering both phases’ crystallinity and conduction band position. The anatase-rich nanoparticles’ higher photocatalytic activity than that of anatase nanoparticles is described by a synergistic effect between anatase and brookite [10,11,59]. According to Tarek et al., there is a higher transformation rate of anatase to rutile than brookite to rutile. The case of anatase-rich or brookite-rich nanoparticles is explained by the increase in contact sites between the anatase nanoparticles, resulting from their aggregation, which act as sites for the rutile nucleation [60].

Until 2001, it was still believed that TiO_2_ in the anatase form appears to be the most efficient semiconductor for environmental applications [3]. Reports suggest that a mixture of anatase and rutile would be the best combination to achieve maximum photocatalytic efficiency [61]. Several commercial samples of TiO_2_ varying in particle size and purity were studied to determine catalytic activity with Degussa P25 grade (a mixture of 70% anatase and 30% rutile material). Nevertheless, Cihlar et al. have studied anatase-brookite nanoparticles at different phase compositions of biphasic TiO_2_, finding anatase-brookite nanoparticles that contain 36% brookite have higher photocatalytic activity and the maximum hydrogen evolution [62]. Regarding crystallography, high contents of anatase conduce to produce effective TiO_2_ photocatalytic systems, providing a larger surface area and sound diffusion of metal nanoparticles on the surface of anatase [7]. There is a wide range of photo-reactivity within specimens of mixtures containing variable contents of anatase and rutile. Rutile may be active or inactive according to the preparation method. Thus, the originality of the preparation method affects the specimen’s physicochemical properties. Many researchers concluded that rutile is catalytically inactive or a much less active form of TiO_2_ [3,12,53,55,63,64].

### 2.4. TiO_2_ and Non-Ti Compounds; Positive Effect of Heterojunctions on Photoactivity

The fabrication of a heterojunction of semiconductors has been demonstrated to be an effective strategy for promoting charge separation during photocatalytic processes. In this sense, the addition of carboxylic acids, i.e., acetic acid, propanoic acid and butyric acid, during hydrolysis showed surprising results in crystallization behaviour and heterojunction formation [50]. A sol-gel synthesis catalysed with acetic acid of nanocrystalline powder of TiO_2_ confirms the enhancement of the transformation from amorphous to anatase and the growth of the brookite phase after supercritical drying [31] was observed, showing the influences of the nature of peptizing acids (H_2_SO_4_, HNO_3_, and CH_3_COOH) on the formations of the rutile phase and on the photocatalytic efficiency of TiO_2_ nanoparticles [65]. The organic impurities act as active sites to promote the anatase-rutile biphasic system [12]. The use of acetic acid species during the gelation mechanism involves the formation of an ester as an intermediate to metal acetate species. Besides acting as an acid catalyst, acetic acid can influence the kinetics of polycondensation, decreasing the speed of reaction, and consequently influencing the gel’s primary particle growth or causing a narrow particle diameter distribution [66]. The side groups (–NH_2_, –H, –OH, and –SH) offer proper gelation mechanisms with complex interactions, including covalent bond and coordination bond interactions between organic acids and metal ions, which are vital to providing rigid gel networks and preferred photostructures [67]. The glycine amino group gets partially deprotonated during the interaction with the active alkaline centres on the metal surface, while COO- groups interact with its acidic centres [68]. The optimal formation of anatase-brookite depends on the nature of the chelating ligand synchronized to the central titanium and the occurrence of the reactants acting as complexing agents; most synthesis of heterophasic TiO_2_ nanoparticles used complex Ti compounds or in situ donor complexes with Ti, with a donor/Ti molar ratio > 1. The amount of brookite can function instead of acidity in the donor/Ti ratio. Cihlar et al. produced particulate biphasic TiO_2_ xerogels using amino acids like glycine and hydroxycarboxylic acids, such as lactic acid (Figure 3) and citric acid; the specific surface areas obtained ranged from 202 to 292 m^2^/g in the materials aimed photocatalytic applications [8].

## 3. Synthesis of TiO_2_ Photocatalyst: A Brief Summary of Methods from the Literature

### 3.1. Synthesis of TiO_2_ Photocatalyst Containing Heterojunctions

Recently, new methods were introduced to synthesize pure anatase and the biphasic heterojunctions anatase-brookite, i.e., fast hydrothermal microwave heating (HMWH). The authors worked at different reaction temperatures between 100–250 °C. To study the relationship between composition, size, and nanocrystals’ microstructure via their characterization [9], special attention was placed on the crystal facet engineering of TiO_2_ anatase (001) nanocrystals synthesized with TiCl_4_ and featuring mesoscopic void space for photo-splitting of water [69]. These faceted TiO_2_ nanostructures were doped with different metal ions nanoclusters and employed as efficient catalysts [70]. Nowadays, PLD produces a high-phase fraction of brookite (~95%) TiO_2_ thin films. The study obtained brookite, anatase, and rutile and evaluated the role of Na incorporation as a mechanism for brookite stabilization in thin films. The results suggest that film phase selection is directed by a single nucleation event, which can be controlled through films processing and structure. The hydrolysis–condensation process demonstrated that chloride ions play a fundamental role in stabilizing the brookite phase and avoiding the recrystallization of brookite to rutile during ageing. The pH of the solution can regulate the effect; a rise in pH leads the Ti complex to donate the organic ligands by this means, allowing edge-shared bonding to form anatase [54]. Via the hydrothermal reaction at 200 °C for, 6 h, from TTIP and acetic acid as precursors produce anatase, while pure rutile appears at 200 °C, 8 h and brookite was revealed at 175 °C and, 7 h with 3M of HCl [71]. HCl promotes the hydrothermal treatment of amorphous anatase-brookite biphasic systems. The tendency of anatase-brookite TiO_2_ was the highest at pH 0.5. In contrast, anatase–rutile was observed using a base NaCl; the ionic strength interrupted the content of brookite. At pH 3, the anatase–brookite system appears after refluxing, and further hydrothermal aging with the addition of salt does not have any influence in the final phase composition [51]. Refluxing temperatures of 83 °C, 15 h led to the formation of the brookite phase. A lower refluxing temperatures of 70 °C increase the biphasic system of anatase-rutile, and higher refluxing temperature of 100 °C provides the heterophase system brookite-rutile [20].

Several authors have widely studied the effect of size on the phase transition sequence of TiO_2_ nanoparticles, and they observed that anatase, brookite, and rutile phases were stable with particle sizes less than 4.9 nm, between 4.9 and 30 nm, and above 30 nm, respectively [10,12,54]. They conclude that even a small amount of brookite can strongly influence the Raman spectra of anatase-brookite samples, giving a theoretical base of information when characterising crystalline phases in nanomaterials [20,72]. The experiments show that the phase transformation among the three polymorphs of TiO_2_ anatase, brookite, and rutile is size-dependent. This is consistent with thermodynamic calculations predicting that air anatase is most stable below 11 nm, brookite between 11 and 35 nm, and rutile above 35 nm [51]. Fewer impurities or surface defects are capable of producing a defined structure to stabilize the size of anatase in a medium [28]. A lower ageing temperature produces an excellent grain size of 6 nm. Due to decreasing, the particle size produces extra edges and vertexes and, therefore, more equivalently unsaturated surface sites that can participate actively in the photoreactions [73].

### 3.2. Complex Synthesis-Principle: Sol-Gel Route

During the sol-gel process, a precursor solution with a metal alkoxide of titania (Ti-OR)_4_ is converted into a solid through the inorganic polymerization network of an aqueous solution, generating a metal oxo polymer network n(Ti-O-Ti) a complete polymerization and loss of solvent lead to the transformation of the liquid sol into a solid gel phase [74]. This method constitutes a hydrolysis based on the nucleophilic substitution of hydroxyl groups by the alkoxide precursor and catalysed by an acid. Second, condensation, with the elimination of water or alcohol through the cross-linking reactions occurring between M–OH groups or between M-OH and M-OR groups [54] (Figure 4). The hydrolysis reactions form the original nuclei or the basic unit of TiO_2_, and the growth of the cross-linked network system of original basic units is led by condensation. The faster sol-gel reaction leads to the formation of amorphous and non-uniform particles, whereas a slow rate has the potential to produce an ordered structure [7]. The presence of water in the reaction media catalyses the crystallization through the arrangement of [TiO_6_] octahedral bridging and aligning them for dehydration and correct crystallization. As a result, a post-heat treatment would not be necessary for the desired crystal structure in the final product [5]. The pH value is a decisive parameter for the relative rates of hydrolysis and condensation of metal alkoxides; edge-shared bonding for anatase crystallization is supported by high pH. High acidity medium, edge-shared bonding is suppressed to some extent, and corner-shared bonding is facile to form rutile [75]. Brookite has both shared edges and corners and is midway between anatase and rutile in terms of edge-shared bonding. This might be attributable to brookite stabilization at intermediate pH [76].

The sol-gel procedure is simple and does not require expensive equipment. The mesoporous structures obtained using this method are stable and have large specific surface areas (hundreds of m^2^/g), and the synthesis uses lower temperatures different from the calcination process [35]. This method also effectively controls particle size, shape, and other properties [77]. Moreover, there is the possibility of designing the material structure and property through the selection of proper sol-gel precursors and other building blocks [51].

#### Low-Temperature- Hydrothermal Synthesis (LTHT)

In hydrothermal conditions, water adsorbs on amorphous titania, modifying [TiO_6_] octahedron to its crystallization [8,78]. The investigations proposed using temperatures of 40 and 50 °C [79], 80 °C [19,80,81]; and, 60 °C and 90 °C [82] to obtain different TiO_2_ heterojunctions.

In low-temperature synthesis, hydrolysis-condensation and crystallization reactions are typically affected by the aqueous solution’s pH value, the solvent’s nature, and the system’s temperature [35]. Phase stability usually depends on the surface environment as a variable [83]; the surface charge can wield an essential management on nanoparticle shape even at room temperature and its interfacial energy modification [76]. The narrow energy stability frame for brookite shifts the selectivity of the nanocrystal nucleation and growth phase, limiting the reaction conditions more than those necessary for anatase or rutile [71].

Phase transformation in hydrothermal synthesis occurs via a dissolution-precipitation pathway. This consists of dissolution due to surface protonation (anatase to rutile) of one phase or hydroxylation (brookite to anatase), followed by precipitation of a more stable colloidal system with heterophase junctions [54]. Aggregation during synthesis involves complicated processes such as chemical reactions, nucleation and growth (I. nuclei into seeds, II seeds into nanocrystals, and III. Precipitation), which often makes it challenging to study the actual aggregation mechanism in chemical reactions [84] (Figure 5). This dynamic interplay of growth and dissolution dictates seeds’ evolution into nanocrystals [85]. Controlling the aggregation of nanoparticles during synthesis is very complicated. However, sol-gel synthesis provides a number of possibilities to organize the primary nanoparticles and results in nanostructures with tailored properties.

Suitable parameters, i.e., type of precursor, T, pH, and donor/Ti rates during the hydrothermal colloidal preparation, significantly affect the gel’s size, distribution, and stability regarding its photocatalytic applications [86].

### 3.3. Acid-Base Donors’ Effect on Complexing Gelation

One way to enhance crystallization at a low temperature is to slow the condensation reaction. That occurs using acidic synthesis solutions, typically preferred in the low-temperature synthesis of TiO_2_ [20]. This leads to a polymeric gel made from chains with few branches [87]. Small oligomers weakly cross-link the polymer with reactive Ti–OH groups formed simultaneously, which quickly lose their shape and tend to crush. Therefore, acid-catalysed synthesis produces denser aerogels with tiny pores [88]. Proton donors favour hydrolytic reactions under acidic conditions. The oxygen atoms can be attacked due to their partial negative charge [74]. In theory, by adjusting the initial pH to a specific precursor, the porosity of the aerogels can be tailored over the rates balanced between the hydrolysis and condensation reactions [89].

Unlike in an alkaline medium, where the condensation rate accelerates hydrolysis, increasing the relative number of protons (H+) generates more positive charges in the OR groups attached to the metal, promoting the hydrolysis rate. Due to the metal ions being positively charged, they repel the protonated OR groups and become linked to water [33]. Then, the hydrolysed types are readily used up into bigger and denser colloidal particles, yielding lighter aerogels with higher pore volumes.

The acid catalysts enhance the transformation of amorphous gels into TiO_2_ with further hydrolysis polycondensation reactions accompanied by structural rearrangements. However, this may not prompt the complete reactions of residual alkoxyl/hydroxyl groups. A complete reaction of all carboxylate-modifying ligands from alkoxide-derived sol-gel materials is difficult to achieve [78]. chemical modification of alkoxides technique is widely used to control hydrolysis and condensation in polymeric sol-gel systems; nevertheless, there are limited reports for particulate sol-gel systems with agents such as carboxylic acids [89]. Biphasic-TiO_2_ nanoparticles containing anatase/brookite heterophase junctions were prepared via low-temperature hydrolysis and polycondensation sol-gel synthesis of titanium isopropoxide in the presence of monochloracetic acid (MCAA) [80]. The stability of metal complexes and their differences with a pattern of complex formation becomes relevant. The use of an acid compound as a chelating agent structure permits the attachment of two or more donor atoms (or sites) to the same metal ion in (Ti^4+^) and simultaneously produces one or more rings [33] (Figure 3A,B).

#### 3.3.1. Aging Effects on Condensation

A continuous cross-linking process occurs after the gelling stage has been reached, and sol-gel chemical reactions still exist during ageing. The liquid inside the pores contains small particles with unreacted sites (sharing OH-), which can be condensed [90]. Additionally, migratory monomers will join the network as an additional condensation process into well-prone areas of secondary nanoparticles such as pores in between its necks, as well as the Oswald ripening process. Figure 4 shows how condensation causes a syneresis as a dimensional change, leading to the contraction of the polymeric network and drawing the remaining solvent from pores [89].

#### 3.3.2. Post Heat Treatments

The modifications in the structure and properties of gels that happen occur during the ageing process have a significant impact on sintering preparation. Anatase crystallinity is upgraded with an increase in the boiling ageing time [54]. Anatase to rutile formation moves to higher temperatures with expanding maturation time. Conversely, the test without ageing reached critical sizes faster and experienced quicker crystallite change. Koparde and Cummings (2008) found that the critical sintering agglomerate changes to rutile. In differentiation, the sintering of amorphous anatase creates brookite agglomerates [91]. In the sol-gel synthesis of TiO_2_, the obtained powder or coating is commonly amorphous. For the desired crystal structure in the product, heat treatment is necessary. The crystal structures of TiO_2_ most widely used in photocatalytic applications of are anatase, rutile, and their mixtures [12,71]. The crystallization of anatase requires a heat treatment like calcination of about 400 °C when it is used in a sol-gel route. In contrast, rutile crystallisation has been reported to occur in the temperature range from 400–1200 °C [50,61].

The phase transition from anatase to brookite occurs during calcination temperatures below or equal to 600 °C due to the thermodynamic stability of brookite under sol-gel conditions with crystallite sizes of 19 nm and up to 50 nm. At higher temperatures, i.e., exceeding or equal to 700 °C, rutile transforms from anatase and brookite [7,27,92]. Bhave et al. prepared xerogels during sol-gel ambient conditions with the dominant phase as brookite presented in samples calcinated at 200 to 500 °C. Tinoco Navarro et al. studied the impact of TiO_2_ calcination within the temperature range of 100–500 °C, it were evident in the alterations in phase and chemical composition, specific surface area, pore size distribution, and band gap energy. These changes were mirrored in the photocatalytic performance of the samples. With the rise in calcination temperature from 100 to 500 °C, there was a marginal increase in anatase content to 81.8%, coupled with a decrease in brookite content [80]. Rutile phase content increases at 300 °C calcination temperature. The rutile phase dominates at 500 °C, and there is a complete transition to the rutile phase at 600 °C. The transformation is directly from the brookite phase to the rutile phase, with no presence of the anatase phase [23].

#### 3.3.3. Heterojunctions-Grain Boundaries

Equal and unequal-sized measured sintering phases indicate that atoms in the grain boundary are essential for phase transformation activation. This phenomenon explains that the grain boundary region, as a planar zone filled with atoms, not in ideal crystallite arrangement and with weak bonding, withdraws favoured sites for nucleation and growth of a new phase [84]. The study of sintering-induced phase transformation of TiO_2_ by Mao et al. (2015) found that atoms in the brookite phase begin to nucleate and increase at the grain boundary. The core regions of the original anatase nanoparticles then melt because of the rising temperature during sintering. The newly generated brookite phase subsequently develops into the core regions in stages, forming a larger nanoparticle with an ordered atomic arrangement. Initially, atoms from the amorphous nanoparticle nucleate to the anatase layer by layer and spread onto the surface of the larger core−shell nanoparticle [93]. The grain boundary is shaped among the original core crystal and the newly formed anatase lattice. The new anatase lattice assists the next phase transformation from anatase to brookite [76]. The grain boundaries created during sintering prompt the consecutive nucleation and the growth of a new phase, and the initial structures of two nanoparticles, such as core−shell or amorphous structure, considerably influence the dynamics of the sintering-induced phase transformation.

### 3.4. Drying Techniques Obtention of an Aerogel or Xerogel Catalyst

The wet gels may be dried to produce the so-called xerogels or aerogels. There are several techniques, namely, supercritical drying (SCD), freeze-drying, ambient pressure drying (APD), and, recently, organic solvent sublimation drying (OSSD) [94]. The drying of the gels dictates their pore dimensions and overall textural properties; thus, the chosen method is highly relevant. Doped TiO_2_ aerogels with phase areas over 200 m^2^/g have been produced through supercritical drying and successfully applied for photocatalytic reformation of ethanol and hydrogen production with a rate of 7.2 mmol.h^−1^.g^−1^ [95].

Conventional drying techniques like simple heating or drying at room temperature lead to shrinkage of the gel network, and the rough surfaces will collapse upon drying due to these capillary forces acting on pore walls and producing xerogels. When the gel network is formed by hydrolysis and condensation, the solvents used for synthesis occupy the pores. So, removing the pore liquid without shrinking the gel network is crucial due to the formation of liquid–vapour meniscus, which recedes the pore walls [96].

## 4. Mechanisms of Photocatalytic Activity

The photo-activity occurs over the photocatalyst surface. Incident photons arrive with enough energy (hv > Ebg), producing excitation of electrons from the valence band (VB) to the conduction band (CB). The valence band’s positive holes (h+) react with the hydroxylated surface to produce OH radicals, considered the most potent oxidizing agents leading advanced oxidation processes (AOPs) of chemical purification treatment of organic material in water [3]. TiO_2_ photodecomposition reactions start when conduction band electrons react with surface-adsorbed oxygen to yield superoxide radical anions ($O_{2}^{-∎}$), while valence band holes react directly with surface-adsorbed pollutant molecules or with surface-adsorbed water molecules (or hydroxyl groups) to produce surface-bound OH radicals, [97] (Figure 6 steps 1–3).

The hydrogen reduction is generated by the redox reaction on the surface of the catalyst through the electron reaction with H^+^ and the oxidation of a hole with H_2_O to form oxygen. The photocatalytic water splitting occurs as it is presented in the chemical Equations (1) and (3) (Figure 7). However, for a more efficient H_2_ using visible light active semiconductor, the band gap should be in the range of 1.23–3 eV The reaction kinetics would involve determining the rate expressions for each of these steps, considering factors like the concentration of reactants and the intensity of light. The rate-limiting steps are typically associated with the reduction of protons (H^+^) to form hydrogen gas (H_2_). This reduction step involves the transfer of electrons to protons, resulting in the evolution of hydrogen. The reduction of protons to produce hydrogen gas is often considered the rate-limiting step in photocatalytic water splitting using TiO_2_. This step is influenced by various factors, including the availability of electrons and protons, as well as the efficiency of charge carrier separation and migration on the TiO_2_ surface [17].
2e^−^ + 2H^+^ → H_2_(1)
4h^+^ + 2H_2_O + → O_2_ + 4H^+^(2)
2H_2_O(l) → H_2_(g) + 1/2O_2_(g)(3)

### 4.1. Influence of Physicochemical Properties on Charge Diffusion

The physicochemical properties benefits offered by aerogel nanostructures are imperative for energy conversion and storage applications [32,99,100]. The large surface area of the nanomaterial promotes molecular adsorption, and photoreactions at the solid-liquid interface are enhanced. The open porous structure facilitates ionic mobility and electrolyte diffusion for fabricating high-performance energy conversion. For instance, the adsorption of large dye molecules on the semiconductor surface and electrolyte diffusion at the interface is significant for dye-sensitized solar cells [86].

The probability of the charge carriers (e^−^/h^+^) reaching the photocatalyst surface without being captured by the crystal defects increases with improved crystallinity [14,101]. In the case of biphasic TiO_2_ aerogels, rutile tends to result in relatively low surface area material due to the formation of compact aggregates [93]. On the other hand, anatase and brookite nanomaterials consist of porous aggregates and are suitable for application in photocatalysis, sensors, and dye-sensitized solar cells with higher surface areas over 200 m^2^/gr [82].

Particle size and crystallinity of the semiconductor control the charge carrier’s migration rate. A smaller particle size is expected to lead to better titanium dioxide solubility, which is consistent with the increase in the rate of growth due to coarsening. The greater surface charge is expected to lead to better electrostatic repulsions between particles, which is consistent with the decrease in the growth rate due to oriented aggregation [51]. The size quantization effect of the TiO_2_ nanoparticles benefits the photocatalytic performance, and the bandgap widens as reflected by the blue shift in the absorption edge. The shift of (CB) and (VB) towards more negative and positive potentials, respectively, favour the redox process that is not occurring in their correlative aggregates [55].

In the case of water splitting, the electronic arrangement, light absorption capability, and charge transport phenomena have a primary role in applying TiO_2_ as a photocatalyst. The valence band (VB) potential must be more positive than the oxidation potential of water (1.23 V), and the conduction band (CB) potential must be more negative than the reduction potential of water (0.0 V) for the occurrence of the photocatalytic reactions [53].

### 4.2. Recombination of Charge Carriers during Photocatalytic Performance

A wide range of applications connected with each crystalline form of titania and the lattice structure differences between heterojunctions lead to the physicochemical properties being optimized to control the photoelectronic structure and bulk diffusion ability of charge carriers [54]. Charge carriers (e^–^ + h^+^) can also recombine, dissipating energy. Recombination may occur in bulk or on the surface of the photocatalyst, being facilitated by defects and impurities in the structure for instance, in one theory for a biphasic system anatase-rutile, anatase scavenges the electrons photo-produced in rutile [102]. It stabilizes the charge separation, preventing rapid recombination [61]. The band gap width for anatase is approximately 3.2 eV (l = 388 nm), for rutile, it is 3.0 eV (l = 414 nm) and for brookite, it is 3.2 eV (l = 388 nm). However, band gaps depend on the measurement method, temperature, and the lattice constant [5]. With the high refractive index, anatase is preferred mainly in photo-catalysis and photovoltaics. The exceptional light scattering efficiency and UV absorptivity of rutile enables its utility as a filter in solar creams, pigments, opacifiers, and optical communication devices (isolators, modulators, and switches). Although, brookite is a rare polymorphs in nature and is synthesized only under sensitive conditions, its importance in photocatalysis, photovoltaics and lithium ion insertion has been recently realized [103].

### 4.3. Presence of a Scavenger in Photo-Oxidation/Reduction Mechanism

A scavenger is a substance added to a chemical reaction or mixture in order to remove or inactivate the effect of impurities and unwanted reaction products from another donor. The donors researched so far are both organic and inorganic species such as ethanol [44], methanol [104], propanol [105], lactic acid [8], and formic acid [16]. All these donors are very effective in improving hydrogen productivity [27]. Figure 5 shows the participation of methanol as a sacrificial agent. However, some researchers have recently reported on the halogen-based sacrificial donor systems Lei Huang [106] demonstrated that Clˉ can be readily adsorbed on the surface of titania during a study of pure water splitting with Cl-TiO_2_ prepared using the hydrothermal method. Girivyankatesh et al. evaluated the effect of available chloride ions as a sacrificial donor and received a positive impact on the photo-hydrogen generation reaction. This was attributed to the ability of chloride ion scavenging holes with favourable implications on the hydrogen evolution rate [107].

The absence of suitable electron and hole scavengers dissipated the stored energy within a few nanoseconds via recombination. If a suitable sacrificial agent or a surface defect state is available to trap the electron or hole, their recombination is prevented, and a subsequent redox reaction may occur [3]. These charge carriers induce the reduction or oxidation of species adsorbed on the surface of the metal-oxide photocatalyst nanoparticles. The trapping experiments also broadly study the photo-degradation of dye pollutants using various aerogels [108].

### 4.4. TiO_2_ Doping Techniques Improvement on Photo Efficiency

Various reports of successful photocatalysts for H_2_ production employed several techniques that influence hydrogen evolution (Figure 8). To date, TiO_2_ is still considered one of the most promising photocatalysts for the mentioned application.

Doping is a common approach to improve the photo-response of TiO_2_ in both UV and visible light regions. Some of these methods include metal and non-metal doping, maximizing its photocatalytic performance [63]. During the process, the doped ions are incorporated into the bulk TiO_2_ aerogels or form clusters of mononuclear complexes highly dispersed on the surface [44,109].

#### 4.4.1. Non-Metallic Doping

Non-metal doping, including with carbon (C), nitrogen (N), fluorine (F), and Sulphur (S) of titanium dioxide (TiO_2_), significantly enhances its photocatalytic activity under visible light, offering a promising solution for improved performance in photocatalysis applications [34]. Carbon doping, using elemental carbon, permeates the lattice of TiO_2_, substituting the lattice of an oxygen atom to form an O-Ti-C bond. It has several advantages: metallic conductivity, extensive electron storage capacity, as an acceptor of the photon-excited electrons, and a wide range of visible light absorption (400–800 nm), facilitating charge transfer from the bulk TiO_2_ to the surface region during the oxidation process [88,110,111].

Nitrogen doping provides small ionization energy and high stability when, introduced into the TiO_2_ structure. In N-doped TiO_2_, the lattice oxygen atom is substituted to form nitride (Ti-N) or oxynitride (O-Ti-N) due to arrangements of elemental nitrogen permeating the lattice of TiO_2_—some of the reasons why N-doping mechanisms expand the light absorption and photocatalytic efficiency of TiO_2_ [1]. The generated oxygen vacancies, stabilized by nitrogen due to charge compensation, might act as colour centres, imparting a visible light response, and nitrogen is bound to hydrogen as NHx species in interstitial sites [14]. Nitrogen in TiO_2_ aerogels increases visible light activity and creates a synergetic effect between Pt and nitrogen in TiO_2_, which was observed in samples with higher nitrogen for water splitting reduction [90].

Fluorine doping in TiO_2_ inhibits the anatase-to-rutile phase transformation, ensuring a highly crystalline anatase structure even at 700 °C. Simultaneously, it fosters the creation of surface defects, contributing to enhanced photoactive absorption around 365 nm [112], with the effect intensifying at higher calcination temperatures [113]. Cihar et al. introduced fluorine (F) and chlorine (Cl) ions by using substitutive acetic acids (SAAs) as precursors during hydrothermal sol-gel synthesis [19].

Sulphur doping in titanium dioxide (TiO_2_) through cationic or anionic routes brings about distinct modifications that enhance its photoactivity. In cationic doping, sulphur replaces titanium ions in the TiO_2_ lattice, typically with S^4+^ or S^6+^ ions. On the other hand, anionic doping involves sulphur substituting oxygen ions in the TiO_2_ structure with S^2−^ ions. This substitution also affects the electronic band structure, narrowing the band gap and allowing efficient utilization of visible light in photocatalytic processes [114].

#### 4.4.2. Metallic Doping

In exploring the impact of transition metal ion doping on titanium dioxide (TiO_2_), the studies have investigated the effects of Fe, Co, Ni, and Cu dopants. The focus is on understanding their influence on charge trapping, recombination dynamics, interfacial transfer processes, and photocatalytic activity within the TiO_2_ framework.

Using Cu^2+^ as a dopant of TiO_2_ catalyst increased the photocatalytic degradation of acid orange 7 (AO7) [109] and tartrazine [45]. The reason is that Cu^2+^ acts as a scavenger of electrons to form Cu^+^, enhancing the oxidation of the substrate [44,104].

Fe is another worthy dopant for increasing the photocatalytic performance of TiO_2_ [6,110]. Fe^3+^ can be incorporated into the TiO_2_ lattice due to its anionic radius (0.64 Å) being close to that of Ti^4+^ (0.68 Å). It has been demonstrated that Fe dopant can facilitate the separation of photogenerated electrons and holes [103]. Fe^3+^ also traps photo-generated holes to form Fe^4+^, which reacts with the surface-adsorbed hydroxyl ions to produce hydroxyl radicals and O_2_ in the surface lattice. Thereby affecting the photocatalytic activity [103,115]. Bharti et al. studied the moderate doping of Fe and Co elements using air plasma treatment, which slightly decreases the band gap, and induces a significant shift in optical properties. This change is attributed to the emergence of Ti^3+^ and oxygen vacancies within the band gap, highlighting a substantial influence on the material’s optical behaviour [116].

Nickel (Ni) doping in titanium dioxide (TiO_2_) introduces defect states within the band-gap and enhances intra-band states through interaction with intrinsic oxygen vacancies [117]. This doping results in improved adsorption in the visible region and increased photocatalytic degradation rates under visible light, highlighting its potential for enhancing TiO_2_ performance in visible light-driven applications [118]. Tinoco Navarro et al. found that Ni was homogeneously distributed in TiO_2_ lattice of aerogel nanoparticles [109], and, presented the highest photocatalytic activity among the other dopants namely Fe, Co and Cu.

High photocatalytic efficiency is achieved through with novel metals such as Au, Ag, Pt, Pd, and Rh nanoparticles, which have been reported as very efficient dopants for the visible-light activation of TiO_2_ photocatalyst [7,39]. The effective transfer of the photogenerated electrons from the conduction band (CB) of TiO_2_ to the metal particles is associated with the Fermi levels of these noble metals, which are lower than that of TiO_2_ [16,35]. Stronger photocatalytic reactions are produced by the electron trapping process, which significantly reduces the electron-hole recombination rate [35].

H_2_ production efficiency of bimetallic Cu-Pt TiO_2_ was evaluated using glycerol as a scavenger in a water mixture under UV–vis light irradiation as proposed by [119]. The superior efficiency of Cu-Pt/TiO_2_ is ascribed to high electron density on the bimetallic particles (due to the interaction of Cu and Pt) when compared to the single metal (Cu or Pt) doped TiO_2_. The different morphologies and photocatalytic performance of TiO_2_ composites or metal doping have already been studied, where between them, the most efficient catalyst was still Pt nanoparticles, producing about 181,770 mmol.h^−1^.gr^−1^ [15,19,43,44,106,120]. Table 1 presented the advantages and limitations of different doping methods.

#### 4.4.3. Doping TiO_2_ with Rare Earth Elements

In recent efforts to enhance the photocatalytic properties of titanium dioxide (TiO_2_), the strategy of doping with rare earth elements has emerged as a promising avenue. Rare earth elements, including cerium (Ce) [125], lanthanum (La) [126], and neodymium (Nd) [127], exhibit unique electronic configurations that can influence the electronic structure and surface properties of TiO_2_ [128]. Doping TiO_2_ with rare earth elements has been shown to enhance light absorption in the visible region, thereby extending its photoresponse range. The introduction of rare earth elements can also improve charge carrier separation and migration, mitigating the recombination of electron-hole pairs, which is a common limitation in traditional TiO_2_ photocatalysis [129].

Several studies have reported the positive impact of rare earth element doping on TiO_2_ photocatalysis. For example, Ce-doped TiO_2_ nanomaterials have demonstrated improved photocatalytic activity in the degradation of organic pollutants under visible light irradiation [130]. Similarly, La-doped TiO_2_ photocatalysts have exhibited enhanced efficiency in the reduction of carbon dioxide (CO_2_) to valuable hydrocarbons [131]. The incorporation of rare earth elements into TiO_2_ introduces unique electronic states and defect structures, influencing the material’s surface chemistry and catalytic performance [14].

In conclusion, the doping of TiO_2_ with rare earth elements represents an evolving and promising strategy to tailor the photocatalytic properties of TiO_2_ for improved performance in various applications.

### 4.5. Doping with Metals the Role in Enhancement the Photocatalytic Activity of TiO_2_

Metal doping exerts a multifaceted influence on the photoactivity of TiO_2_, shaped by various characteristics of the dopant. These include (i) concentration and distribution within the lattice, (ii) energy level positioning within the lattice, (iii) d electron configuration, and (iv) electron donor density, all interacting with incident light intensity. The intricate interplay of these factors underscores the complex relationship between metal doping and the resulting photoactive properties of TiO_2_ [117]. Doping titanium dioxide (TiO_2_) with metals has emerged as a highly effective strategy for augmenting its photocatalytic activity. In this process, certain metal elements are intentionally introduced into the TiO_2_ lattice, imparting unique properties that significantly enhance its performance. Metal doping plays a crucial role in modifying the electronic structure of TiO_2_, leading to an expanded absorption spectrum that extends into the visible range. This is particularly advantageous as it allows for increased utilization of solar radiation, thereby enhancing the overall efficiency of the photocatalyst [49,132,133]. Additionally, metal-doped TiO_2_ exhibits improved charge carrier separation and migration, mitigating the recombination of photoexcited electrons and holes [14,134]. The presence of metal dopants acts as catalytic sites, facilitating the activation of specific reactions and promoting the generation of reactive oxygen species, which play a key role in pollutant degradation [6,128,130]. Moreover, metal-doped TiO_2_ often demonstrates enhanced stability and longevity, contributing to its applicability in various environmental remediation and energy conversion processes. The versatility of metal doping positions it as a promising avenue for tailoring TiO_2_′s properties, making it a highly efficient and adaptable photocatalyst.

## 5. TiO_2_ Photocatalyst Applications in Energy, Environment, Biomedicine Fields

### 5.1. Transformative Uses of Biphasic TiO_2_ Heterojunctions in Photocatalysis, Energy Storage, Sensors, and Catalysis

Biphasic TiO_2_ heterojunctions, characterized by the presence of distinct crystal phases and composite structures, have emerged as versatile materials with wide-ranging applications across various fields. One notable application lies in the realm of photocatalysis, where these heterojunctions exhibit enhanced efficiency in harnessing solar energy for environmental remediation and water purification [80,135]. The crystal phase junctions within biphasic TiO_2_ enable tailored electronic band structures, promoting improved charge separation and migration during photocatalytic processes.

In the field of energy storage, the development of biphasic TiO_2_ heterojunctions has led to advancements in lithium-ion batteries [136,137]. The unique combination of crystal phases enhances the material’s conductivity and charge storage capacity, contributing to the creation of high-performance electrodes. This innovation is pivotal for the evolution of energy storage technologies, addressing the increasing demand for more efficient and sustainable energy solutions [138].

Furthermore, biphasic TiO_2_ heterojunctions find applications in sensor technology. The controlled integration of different crystal phases allows for the manipulation of the material’s sensing properties, making it highly responsive to changes in the surrounding environment. This capability is harnessed in the creation of sensitive and selective sensors for detecting gases, pollutants, and biomolecules, thereby affecting the fields of environmental monitoring and healthcare diagnostics. Various studies in the field of sensing have yielded promising results. The WO_3_:TiO_2_ (1:6) nanocomposite demonstrated electrochemical sensor capabilities for lead ion detection and effective dye degradation, with commendable capacitance values. This suggests potential applications in environmental water samples for meeting stringent control measures [139]. Additionally, a novel CuO/TiO_2_ heterojunctioned nanointerface, developed using a scalable reactive magnetron sputtering technique, exhibited impressive sensitivity to N_2_O gas at room temperature, surpassing TiO_2_ mono-layer performance. The study emphasizes the importance of ultra-low concentration detection [140]. Furthermore, investigations into H_2_ and CO adsorption on NiO-TiO_2_ heterojunctions revealed enhanced selectivity, providing valuable insights for improving MOS-based gas sensors [139]. Another notable contribution involves a two-layer TiO_2_/SnO_2_ heterojunction sensing film, annealed at 400 °C, exhibiting excellent gas sensing properties, particularly a rapid response time and recovery time to 50 ppm ethanol [141] and Gang Li et al. tried the Co_3_O_4_–TiO_2_ porous heterojunction nano sheets as well with the goal of sensing ethanol [142]. These diverse studies collectively contribute to advancements in sensing technology with implications for environmental monitoring and gas detection [143].

In the domain of catalysis, biphasic TiO_2_ heterojunctions play a crucial role in promoting catalytic activities for various chemical transformations. The composite junctions facilitate synergistic interactions between different phases, resulting in improved catalytic efficiency and selectivity such as the case of Mo-doped titania thin films for degradation of trichloroethylene (TCE) [144]. This has implications for industrial processes, such as the production of fine chemicals and the reduction of environmental pollutant [136,145].

The importance of biphasic TiO_2_ heterojunctions in material development cannot be overstated. These structures serve as a platform for tailoring material properties, providing a means to optimize performance in diverse applications. As researchers continue to explore and refine the design of these heterojunctions, their impact is likely to extend further, contributing significantly to the advancement of technologies in energy, environmental, and catalytic applications.

### 5.2. Versatile Healthcare Uses of Biphasic TiO_2_ Heterojunctions in Biomedicine and Beyond

Biphasic TiO_2_ heterojunctions exhibit promising applications in the field of biology and biomedicine, particularly in the creation of active self-disinfecting antiviral surfaces [146]. The photocatalytic properties of TiO_2_ are leveraged to develop surfaces that can harness light energy to generate reactive oxygen species, leading to the inactivation of viruses and other pathogens on contact [147]. This technology holds immense potential for mitigating the spread of infections in healthcare settings, public spaces, and everyday objects.

TiO_2_-based photocatalysis at the interface with biology and biomedicine extends beyond surface disinfection [148]. These heterojunctions are utilized in the design of antibacterial coatings for medical implants and devices, where the controlled release of reactive species can prevent bacterial colonization and biofilm formation [149,150]. This is critical for enhancing the biocompatibility and longevity of implanted medical devices.

In the realm of diagnostics, biphasic TiO_2_ heterojunctions contribute to the development of biosensors with enhanced sensitivity. Functionalizing the surface with specific biomolecules allows for the selective detection of biomarkers, viruses, or other biological entities. This has implications for rapid and accurate disease diagnosis, enabling timely intervention and treatment [151], Bharat Sharma et al. studied TiO_2_-SnO_2_ heterostructures for environmental biomarkers of diabetes which resulted with high selective detection [152].

Moreover, TiO_2_-based nanomaterials find application in drug delivery systems. The unique properties of biphasic TiO_2_, including its biocompatibility and tunable surface chemistry, make it an ideal candidate for encapsulating and delivering therapeutic agents to specific target sites within the body. This controlled drug release can enhance treatment efficacy while minimizing side effects [153]. In regenerative medicine, biphasic TiO_2_ heterojunctions play a role in tissue engineering. By providing a scaffold with tailored properties, these materials support cell adhesion [154], proliferation, and differentiation [155].

The integration of biphasic TiO_2_ heterojunctions into biological and biomedical applications underscores their versatility and potential to address challenges in healthcare and life sciences. As research in this area progresses, these nanomaterials are likely to contribute significantly to the development of innovative solutions for disease prevention, diagnostics, and treatment, ultimately improving the quality of healthcare and biomedicine.

## 6. Conclusions

This article delves into TiO_2_ photocatalyst, spotlighting structural diversity and the pivotal role of anatase-brookite heterophase junctions in enhancing efficiency. It details various synthesis methods, emphasizing the significance of the sol-gel route and low-temperature hydrothermal synthesis. The exploration of intricate mechanisms governing photocatalytic activity, including the influence of physicochemical properties and challenges in charge carrier recombination, offers a comprehensive overview.

Significant advances include the integration of anatase-brookite junctions, displaying their positive impact. The focus on low-temperature hydrothermal synthesis and the sol-gel route highlights evolving synthesis techniques. The discussion on TiO_2_ doping techniques and heterojunctions reveals promising prospects for improving efficiency, with potential energy, environmental and health-related applications.

Looking forward, opportunities lie in refining heterophase junctions for optimized performance and advancing synthesis methods to enhance structural diversity. Addressing challenges in charge carrier recombination and exploring innovative scavenger strategies are crucial for pushing the boundaries of efficiency. Continued research into TiO_2_ doping techniques and heterostructures may unlock new possibilities for sustainable environmental solutions.

## Figures and Tables

**Figure 1 gels-09-00976-f001:**
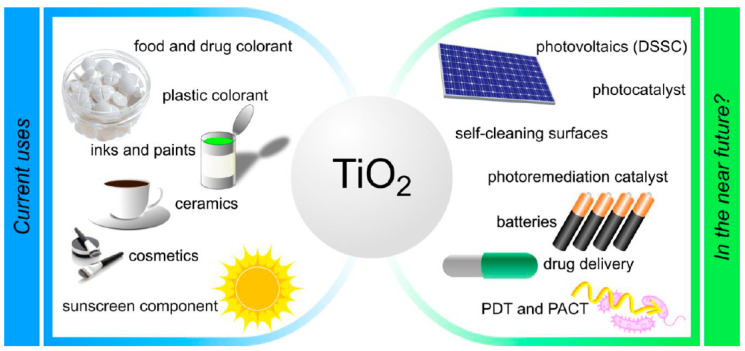
Current applications and future potential use of TiO_2_ [48].

**Figure 2 gels-09-00976-f002:**
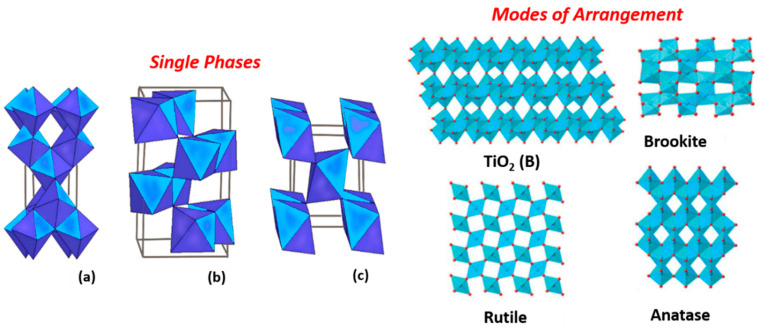
Crystal structures of the (**a**) anatase, (**b**) brookite and (**c**) rutile phase of TiO_2_ [53] and Modes of arrangement of [TiO_6_] octahedrons in titania polymorphs [54].

**Figure 3 gels-09-00976-f003:**
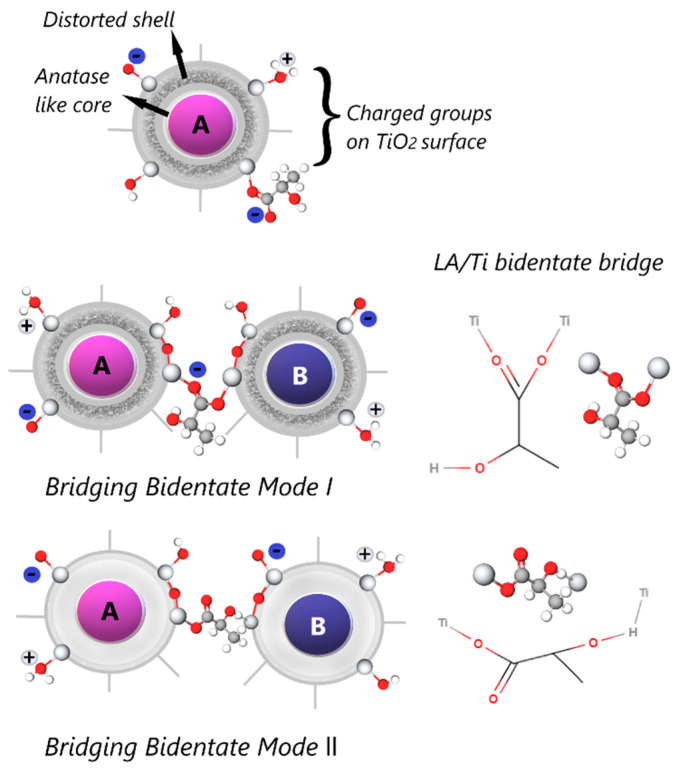
Heterophase junction of anatase-brookite formed by bridging modes from lactic acid.

**Figure 4 gels-09-00976-f004:**
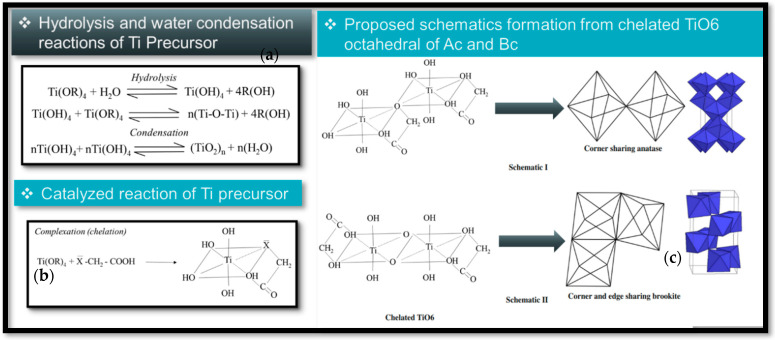
[TiO_6_] octahedral formation via hydrolysis/condensation reactions from the titanium precursor Ti(OR)_4_ during chelate synthesis with carboxylic acid (COOH-) of anatase-brookite. (**a**) Hydrolysis and water condensation reactions of Precursor. (**b**) Catalyzed reaction of Ti precursor. (**c**) Proposed schematics formation from chelated TiO_6_ octahedral of Ac and Bc.

**Figure 5 gels-09-00976-f005:**
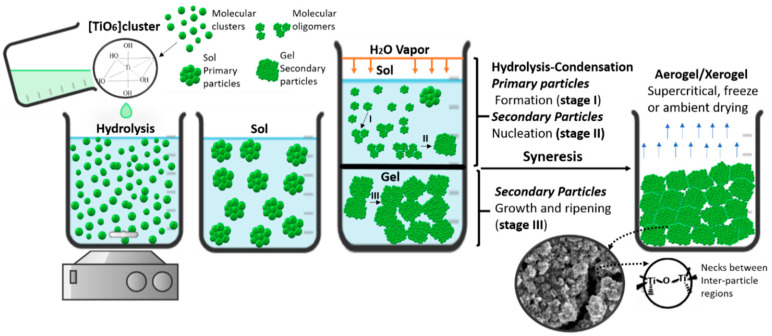
Classical and hydrothermal sol-gel hydrolysis-condensation process of biphasic TiO_2_ aerogel/xerogel.

**Figure 6 gels-09-00976-f006:**
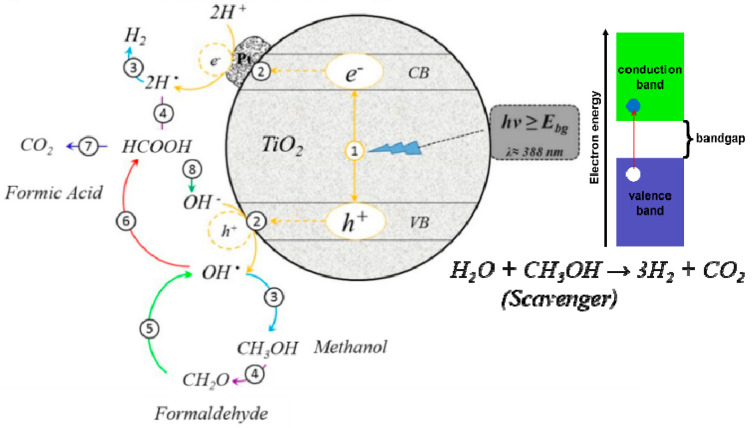
Photocatalytic Mechanism. Schematic picture oxidation/reduction reactions during the photocatalyst excitation during a scavenger’s presence [44].

**Figure 7 gels-09-00976-f007:**
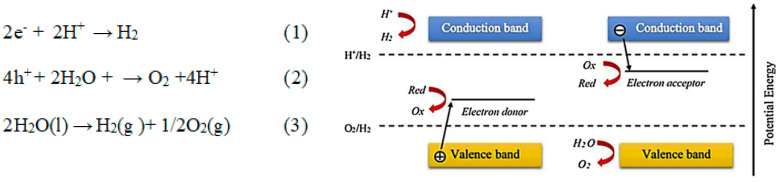
Scheme of hydrogen production by photocatalytic water splitting in the presence of electron donors and acceptors [98].

**Figure 8 gels-09-00976-f008:**
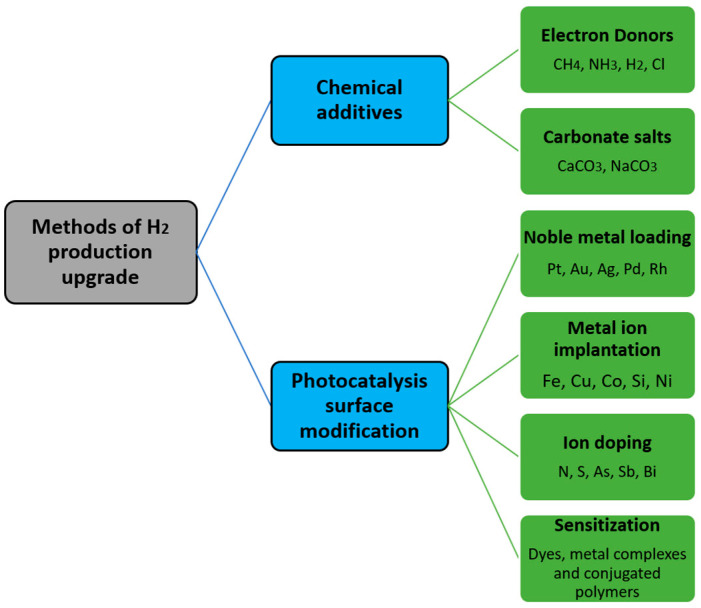
Methods of upgrading H_2_ production [53].

**Table 1 gels-09-00976-t001:** Advantages and limitations of different doping methods.

Doping Methods	Advantages	Limitations
No Metallic Doping (C, N, S, F, Cl) [34,35,118,121]	Enhanced visible light absorption e.g., C, N. Improved charge carrier separation and migration e.g., C, N. Tunable electronic properties e.g., C, F. Altered band structure e.g., Fluorine (F) doping, in particular Potential for improved photo electrochemical performance e.g., Fluorine (F) doping, among others Inherent stability and biocompatibility Cost-effective Minimal risk of introducing impurities	Limited enhancement of TiO_2_ properties. Limited improvement in overall photocatalytic activity. Complexity in the doping process Limited control over vacancy distribution Challenges in precise control of doping Levels Difficulty in achieving uniform doping
Metallic Doping Methods		
Transition Metal Doping (Fe, Co, Ni, Cu) [41,103,109,117]	Enhanced photocatalytic activity Improved charge carrier separation and migration in low concentrations e.g., 0.5, 1 and 5 wt% Tunable electronic band structure and Broadened absorption spectrum Increased visible light responsiveness	Risk of introducing impurities Possibility of reduced biocompatibility Potential toxicity of certain transition metals e.g., Pb, Hg, As, Cd, Fe.
Nobel Metal Doping (Pt, Au, Pd, Ag, Rd, Os) [19,38,122,123,124]	Remarkable enhancement in photocatalytic activity Superior charge carrier separation Excellent stability Enhanced selectivity in certain Photocatalytic reactions Lower susceptibility to charge recombination	Higher cost compared to other doping methods Limited availability of noble metals Potential agglomeration of noble metal nanoparticles Limited impact on visible light absorption

## Data Availability

Not applicable.

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
