# Peer review of "Enhancing Photocatalytic Properties of TiO2 Photocatalyst and Heterojunctions: A Comprehensive Review of the Impact of Biphasic Systems in Aerogels and Xerogels Synthesis, Methods, and Mechanisms for Environmental Applications"

_gels, 2023, doi:10.3390/gels9120976_

Round 1

Reviewer 1 Report

Comments and Suggestions for Authors

This comprehensive review provides a deep dive into the fascinating realm of titanium dioxide (TiO2) photocatalysts, with a strong focus on their structural phases and the impact of heterophase junctions on photocatalytic efficiency. The synthesis methods for TiO2 photocatalysts are thoroughly explored, drawing from a wide range of techniques found in existing literature. Notably, the review delves into the intricacies of complex synthesis principles, including the sol-gel route and its variants like low-temperature hydrothermal synthesis (LTHT), and the fascinating influence of acid-base donors on gelation. 

While the review is undeniably intriguing, we believe that it could benefit from a greater focus on the practical applications of TiO2 photocatalysts and, in particular, heterojunctions. The authors briefly mentioned the capacity to degrade specific chemicals and pollutants, yet it would be valuable to illustrate concrete examples of successful applications to engage the readers further. Furthermore, the absence of information regarding the photodegradation of viruses and bacteria is noteworthy, given the rapid development of this emerging field. Much attention has been devoted to this issue in reviews https://doi.org/10.1016/j.cej.2022.137048 and  https://doi.org/10.1002/cbic.201900229, which are worth mentioning in citations.

Comments on the Quality of English Language

 Minor editing of English language required.

Author Response

Reviewer 1.

This comprehensive review provides a deep dive into the fascinating realm of titanium dioxide (TiO2) photocatalysts, with a strong focus on their structural phases and the impact of heterophase junctions on photocatalytic efficiency. The synthesis methods for TiO2 photocatalysts are thoroughly explored, drawing from a wide range of techniques found in existing literature. Notably, the review delves into the intricacies of complex synthesis principles, including the sol-gel route and its variants like low-temperature hydrothermal synthesis (LTHT), and the fascinating influence of acid-base donors on gelation. 

While the review is undeniably intriguing, we believe that it could benefit from a greater focus on the practical applications of TiO2 photocatalysts and, in particular, heterojunctions. The authors briefly mentioned the capacity to degrade specific chemicals and pollutants, yet it would be valuable to illustrate concrete examples of successful applications to engage the readers further. Furthermore, the absence of information regarding the photodegradation of viruses and bacteria is noteworthy, given the rapid development of this emerging field. Much attention has been devoted to this issue in reviews https://doi.org/10.1016/j.cej.2022.137048 and  https://doi.org/10.1002/cbic.201900229, which are worth mentioning in citations.

Answer.

I appreciate your suggestion. Regarding your enquired. Was created a new section called “TiO2 photocatalyst applications in energy, environment, biomedicine fields” (lines 687-771) and the recommend citations are included with number 158, 160. (see the main document)

  1. TiO2 photocatalyst applications in energy, environment, biomedicine fields

5.1. Transformative Uses of Biphasic TiO2 Heterojunctions in Photocatalysis, Energy Storage, Sensors, and Catalysis

Biphasic TiO2 heterojunctions, characterized by the presence of distinct crystal phases and composite structures, have emerged as versatile materials with wide-ranging applications across various fields. One notable application lies in the realm of photocatalysis, where these heterojunctions exhibit enhanced efficiency in harnessing solar energy for environmental remediation and water purification [93,148]. The crystal phase junctions within biphasic TiO2 enable tailored electronic band structures, promoting improved charge separation and migration during photocatalytic processes.

In the field of energy storage, the development of biphasic TiO2 heterojunctions has led to advancements in lithium-ion batteries [149,150]. The unique combination of crystal phases enhances the material's conductivity and charge storage capacity, contributing to the creation of high-performance electrodes. This innovation is pivotal for the evolution of energy storage technologies, addressing the increasing demand for more efficient and sustainable energy solutions [151].

Furthermore, biphasic TiO2 heterojunctions find applications in sensor technology. The controlled integration of different crystal phases allows for the manipulation of the material's sensing properties, making it highly responsive to changes in the surrounding environment. This capability is harnessed in the creation of sensitive and selective sensors for detecting gases, pollutants, and biomolecules, thereby affecting the fields of environmental monitoring and healthcare diagnostics. Various studies in the field of sensing have yielded promising results. The WO3:TiO2 (1:6) nanocomposite demonstrated electrochemical sensor capabilities for lead ion detection and effective dye degradation, with commendable capacitance values. This suggests potential applications in environmental water samples for meeting stringent control measures[152]. Additionally, a novel CuO/TiO2 heterojunctioned nanointerface, developed using a scalable reactive magnetron sputtering technique, exhibited impressive sensitivity to N2O gas at room temperature, surpassing TiO2 mono-layer performance. The study emphasizes the importance of ultra-low concentration detection [153]. Furthermore, investigations into H2 and CO adsorption on NiO-TiO2 heterojunctions revealed enhanced selectivity, providing valuable insights for improving MOS-based gas sensors [152]. Another notable contribution involves a two-layer TiO2/SnO2 heterojunction sensing film, annealed at 400 °C, exhibiting excellent gas sensing properties, particularly a rapid response time and recovery time to 50 ppm ethanol [154] and Gang Li et al. tried the Co3O4–TiO2 porous heterojunction nano sheets as well with the goal of sensing ethanol [155] . These diverse studies collectively contribute to advancements in sensing technology with implications for environmental monitoring and gas detection [156].

In the domain of catalysis, biphasic TiO2 heterojunctions play a crucial role in promoting catalytic activities for various chemical transformations. The composite junctions facilitate synergistic interactions between different phases, resulting in improved catalytic efficiency and selectivity such as the case of Mo-doped titania thin films for degradation of trichloroethylene (TCE) [157]. This has implications for industrial processes, such as the production of fine chemicals and the reduction of environmental pollutant [149,158].

The importance of biphasic TiO2 heterojunctions in material development cannot be overstated. These structures serve as a platform for tailoring material properties, providing a means to optimize performance in diverse applications. As researchers continue to explore and refine the design of these heterojunctions, their impact is likely to extend further, contributing significantly to the advancement of technologies in energy, environmental, and catalytic applications.

5.2. Versatile healthcare Uses of Biphasic TiO2 Heterojunctions in Biomedicine and beyond

Biphasic TiO2 heterojunctions exhibit promising applications in the field of biology and biomedicine, particularly in the creation of active self-disinfecting antiviral surfaces [159]. The photocatalytic properties of TiO2 are leveraged to develop surfaces that can harness light energy to generate reactive oxygen species, leading to the inactivation of viruses and other pathogens on contact  [160]. This technology holds immense potential for mitigating the spread of infections in healthcare settings, public spaces, and everyday objects.

TiO2-based photocatalysis at the interface with biology and biomedicine extends beyond surface disinfection [161]. These heterojunctions are utilized in the design of antibacterial coatings for medical implants and devices, where the controlled release of reactive species can prevent bacterial colonization and biofilm formation [162,163]. This is critical for enhancing the biocompatibility and longevity of implanted medical devices.

In the realm of diagnostics, biphasic TiO2 heterojunctions contribute to the development of biosensors with enhanced sensitivity. Functionalizing the surface with specific biomolecules allows for the selective detection of biomarkers, viruses, or other biological entities. This has implications for rapid and accurate disease diagnosis, enabling timely intervention and treatment [164], Bharat Sharma et al. studied  TiO2-SnO2 heterostructures for environmental biomarkers of diabetes which resulted with high selective detection [165].

Moreover, TiO2-based nanomaterials find application in drug delivery systems. The unique properties of biphasic TiO2, including its biocompatibility and tunable surface chemistry, make it an ideal candidate for encapsulating and delivering therapeutic agents to specific target sites within the body. This controlled drug release can enhance treatment efficacy while minimizing side effects [166]. In regenerative medicine, biphasic TiO2 heterojunctions play a role in tissue engineering. By providing a scaffold with tailored properties, these materials support cell adhesion [167] , proliferation, and differentiation [168].

The integration of biphasic TiO2 heterojunctions into biological and biomedical applications underscores their versatility and potential to address challenges in healthcare and life sciences. As research in this area progresses, these nanomaterials are likely to contribute significantly to the development of innovative solutions for disease prevention, diagnostics, and treatment, ultimately improving the quality of healthcare and biomedicine.

Reviewer 2 Report

Comments and Suggestions for Authors

Tinoco et al. have summarized the recent advances in TiO2 photocatalysts, especially focusing on the synthesis methods, mechanisms, and environmental applications. The authors start with an introduction about the importance of TiO2 materials as photocatalysts and the benefits of heterojunction structures. The key points in the subsequent sections are appropriately highlighted. The schematic diagrams aid readers greatly in quickly grasping the main ideas. The authors elucidate the photocatalytic mechanisms, emphasize the roles of physical properties, synthesize methods, and impurity doping. The discussed synthesis techniques and future environmental applications are highly relevant for materials scientists and engineers. I found the manuscript technically sound, well-organized, and suitable for publication after minor revisions.

Comments:

(1)  The abstract could be more concise and structured, highlighting only the most important points. Some paragraphs in the introduction may be condensed to avoid repetition with later sections.

(2)  The section on photocatalytic mechanisms could provide more details about the reaction kinetics and rate-limiting steps. The authors could compare the advantages and limitations of different doping methods in a table format.

(3)  The conclusion section could connect back to the introduction and highlight the most significant advances and future opportunities.

(4)  In the references section, please include some recently published articles from 2020 onwards to further update the content.

(5)  Consider adding a separate section to highlight the potential applications of TiO2 photocatalysts in areas like energy, environment, biomedicine etc.

Author Response

Reviewer 2.

Tinoco et al. have summarized the recent advances in TiO2 photocatalysts, especially focusing on the synthesis methods, mechanisms, and environmental applications. The authors start with an introduction about the importance of TiO2 materials as photocatalysts and the benefits of heterojunction structures. The key points in the subsequent sections are appropriately highlighted. The schematic diagrams aid readers greatly in quickly grasping the main ideas. The authors elucidate the photocatalytic mechanisms, emphasize the roles of physical properties, synthesize methods, and impurity doping. The discussed synthesis techniques and future environmental applications are highly relevant for materials scientists and engineers. I found the manuscript technically sound, well-organized, and suitable for publication after minor revisions.

Comments:

  • The abstract could be more concise and structured, highlighting only the most important points. Some paragraphs in the introduction may be condensed to avoid repetition with later sections.

Answer

Thanks a lot for your comment. The Abstract was structured highlighting only the most important points lines 11-19, and in the introduction  were condensed the paragraph that were containing repetitive information (see main document) line 44-53.

  • The section on photocatalytic mechanisms could provide more details about the reaction kinetics and -rate-limiting steps The authors could compare the advantages and limitations of different doping methods in a table format.

Answer

Thanks a lot for your comment. The section was improved (Line 467-476). And we provide a table comparing the advantages and limitations of different doping methods (line 628) and additional new section “4.3 Doping TiO2 with rare earth elements” line 631.

  • The conclusion section could connect back to the introduction and highlight the most significant advances and future opportunities.

Answer

Thanks a lot for your comments. The conclusion where connected with the introduction highlight the most significant advances and future opportunities (see main document) lines 765-783.

  • In the references section, please include some recently published articles from 2020 onwards to further update the content.

Answer

Thanks a lot for your comment. The reference section was updated with most of the documents from 2020 to 2023 (see main document).

  • Consider adding a separate section to highlight the potential applications of TiO2 photocatalysts in areas like energy, environment, biomedicine etc.

Answer

I appreciate your suggestion. Regarding your enquired. Was created a new section named as follows (lines 687-771):

  1. TiO2 photocatalyst applications in energy, environment, biomedicine fields

5.1. Transformative Uses of Biphasic TiO2 Heterojunctions in Photocatalysis, Energy Storage, Sensors, and Catalysis

Biphasic TiO2 heterojunctions, characterized by the presence of distinct crystal phases and composite structures, have emerged as versatile materials with wide-ranging applications across various fields. One notable application lies in the realm of photocatalysis, where these heterojunctions exhibit enhanced efficiency in harnessing solar energy for environmental remediation and water purification [93,148]. The crystal phase junctions within biphasic TiO2 enable tailored electronic band structures, promoting improved charge separation and migration during photocatalytic processes.

In the field of energy storage, the development of biphasic TiO2 heterojunctions has led to advancements in lithium-ion batteries [149,150]. The unique combination of crystal phases enhances the material's conductivity and charge storage capacity, contributing to the creation of high-performance electrodes. This innovation is pivotal for the evolution of energy storage technologies, addressing the increasing demand for more efficient and sustainable energy solutions [151].

Furthermore, biphasic TiO2 heterojunctions find applications in sensor technology. The controlled integration of different crystal phases allows for the manipulation of the material's sensing properties, making it highly responsive to changes in the surrounding environment. This capability is harnessed in the creation of sensitive and selective sensors for detecting gases, pollutants, and biomolecules, thereby affecting the fields of environmental monitoring and healthcare diagnostics. Various studies in the field of sensing have yielded promising results. The WO3:TiO2 (1:6) nanocomposite demonstrated electrochemical sensor capabilities for lead ion detection and effective dye degradation, with commendable capacitance values. This suggests potential applications in environmental water samples for meeting stringent control measures[152]. Additionally, a novel CuO/TiO2 heterojunctioned nanointerface, developed using a scalable reactive magnetron sputtering technique, exhibited impressive sensitivity to N2O gas at room temperature, surpassing TiO2 mono-layer performance. The study emphasizes the importance of ultra-low concentration detection [153]. Furthermore, investigations into H2 and CO adsorption on NiO-TiO2 heterojunctions revealed enhanced selectivity, providing valuable insights for improving MOS-based gas sensors [152]. Another notable contribution involves a two-layer TiO2/SnO2 heterojunction sensing film, annealed at 400 °C, exhibiting excellent gas sensing properties, particularly a rapid response time and recovery time to 50 ppm ethanol [154] and Gang Li et al. tried the Co3O4–TiO2 porous heterojunction nano sheets as well with the goal of sensing ethanol [155] . These diverse studies collectively contribute to advancements in sensing technology with implications for environmental monitoring and gas detection [156].

In the domain of catalysis, biphasic TiO2 heterojunctions play a crucial role in promoting catalytic activities for various chemical transformations. The composite junctions facilitate synergistic interactions between different phases, resulting in improved catalytic efficiency and selectivity such as the case of Mo-doped titania thin films for degradation of trichloroethylene (TCE) [157]. This has implications for industrial processes, such as the production of fine chemicals and the reduction of environmental pollutant [149,158].

The importance of biphasic TiO2 heterojunctions in material development cannot be overstated. These structures serve as a platform for tailoring material properties, providing a means to optimize performance in diverse applications. As researchers continue to explore and refine the design of these heterojunctions, their impact is likely to extend further, contributing significantly to the advancement of technologies in energy, environmental, and catalytic applications.

5.2. Versatile healthcare Uses of Biphasic TiO2 Heterojunctions in Biomedicine and beyond

Biphasic TiO2 heterojunctions exhibit promising applications in the field of biology and biomedicine, particularly in the creation of active self-disinfecting antiviral surfaces [159]. The photocatalytic properties of TiO2 are leveraged to develop surfaces that can harness light energy to generate reactive oxygen species, leading to the inactivation of viruses and other pathogens on contact  [160]. This technology holds immense potential for mitigating the spread of infections in healthcare settings, public spaces, and everyday objects.

TiO2-based photocatalysis at the interface with biology and biomedicine extends beyond surface disinfection [161]. These heterojunctions are utilized in the design of antibacterial coatings for medical implants and devices, where the controlled release of reactive species can prevent bacterial colonization and biofilm formation [162,163]. This is critical for enhancing the biocompatibility and longevity of implanted medical devices.

In the realm of diagnostics, biphasic TiO2 heterojunctions contribute to the development of biosensors with enhanced sensitivity. Functionalizing the surface with specific biomolecules allows for the selective detection of biomarkers, viruses, or other biological entities. This has implications for rapid and accurate disease diagnosis, enabling timely intervention and treatment [164], Bharat Sharma et al. studied  TiO2-SnO2 heterostructures for environmental biomarkers of diabetes which resulted with high selective detection [165].

Moreover, TiO2-based nanomaterials find application in drug delivery systems. The unique properties of biphasic TiO2, including its biocompatibility and tunable surface chemistry, make it an ideal candidate for encapsulating and delivering therapeutic agents to specific target sites within the body. This controlled drug release can enhance treatment efficacy while minimizing side effects [166]. In regenerative medicine, biphasic TiO2 heterojunctions play a role in tissue engineering. By providing a scaffold with tailored properties, these materials support cell adhesion [167] , proliferation, and differentiation [168].

The integration of biphasic TiO2 heterojunctions into biological and biomedical applications underscores their versatility and potential to address challenges in healthcare and life sciences. As research in this area progresses, these nanomaterials are likely to contribute significantly to the development of innovative solutions for disease prevention, diagnostics, and treatment, ultimately improving the quality of healthcare and biomedicine.

Reviewer 3 Report

Comments and Suggestions for Authors

The manuscript is well written and can be accepted for the publication in Gels-Mdpi journal after minor revision.

1-      The authors should explain how they overcome the disadvantages or the limitations of using TiO2 as a Photocatalyst; give us several examples with references in the introduction section.

2-      Page 2, at the first paragraph, after H2 production, the author should write more references.

3-      Page 21, please put the caption of figure 4 under the figure and remove it from the main paragraph.

4-      Insert the line numbers, to facilitate the revision of manuscript.

5-      Rewrite the chemical compounds such as: TiO2, should be written as TiO2 through all the manuscript.

6-      Page 10, the modification should be in capital letters through all the manuscript.

7-      In the last section, doping TiO2 with metals should be written in details, showing the role of doping in enhancement the photocatalytic activity of TiO2.

8-      The author should talk about doping TiO2 with rare earth elements.

9-      Updated the list of references.

Author Response

Reviewer 3.

The manuscript is well written and can be accepted for the publication in Gels-Mdpi journal after minor revision.

1-The authors should explain how they overcome the disadvantages or the limitations of using TiO2 as a Photocatalyst; give us several examples with references in the introduction section.

Answer

I appreciate your suggestion.

The Introduction was complemented with your suggestion (line 84-101).

Recent advancements in overcoming the limitations of titanium dioxide (TiO2) as a photocatalyst involve innovative strategies aimed at enhancing its efficiency. Doping TiO2 with non-metals (e.g., nitrogen)[35] or metals (e.g., silver, copper) remains a promising avenue, modifying its electronic structure and improving charge separation for heightened photocatalytic activity [36]. Surface modifications, such as coating TiO2 with materials like graphene or carbon-based substances, have been explored to enhance surface area and inhibit charge carrier recombination, leading to improved photocatalytic performance. Hybridizing TiO2 with other semiconductor materials, like ZnO[37]  or CdS[23], to form heterojunctions, has been shown to enhance charge separation and overall photocatalytic efficiency. Strategies to extend TiO2's absorption into the visible range, such as bandgap engineering or coupling with narrow-bandgap semiconductors, are being pursued to enhance its photocatalytic activity under solar light [38]. Introducing co-catalysts like Pt or Pd onto TiO2 surfaces improves charge transfer and enhances the efficiency of specific reactions [39,40]. Additionally, developing heterogeneous photocatalysis systems, including immobilizing TiO2 on various supports or incorporating it into membranes, aims to improve catalyst recovery, reusability, and stability [41,42]. These approaches collectively represent the forefront of efforts to address the limitations of TiO2, paving the way for its broader and more effective application in photocatalysis.

2-Page 2, at the first paragraph, after H2 production, the author should write more references.

Answer

Thanks a lot for your comment. The references were added (line 42).

3-Page 21, please put the caption of figure 4 under the figure and remove it from the main paragraph.

Answer

Thanks a lot for your comment. The figure was accommodated with the caption below. (line 236)  

4-Insert the line numbers, to facilitate the revision of manuscript.

Answer

5-Rewrite the chemical compounds such as: TiO2, should be written as TiO2 through all the manuscript.

Answer

Thanks a lot for your comment. The subscripts were arranged in all the manuscript.

6-Page 10, the modification should be in capital letters through all the manuscript.

Answer

Thanks a lot for your comment. The word Modification was updated in in capital letters through all the manuscript.

7-In the last section, doping TiO2 with metals should be written in details, showing the role of doping in enhancement the photocatalytic activity of TiO2.

Answer

Thanks a lot for your comment. It was created own section with this purpose and as well updated the section of non-metal doping (lines 580-592) and metal doping (Lines 607-618) new sections “4.4.3 Doping TiO2 with rare earth elements” (lines 639-661). And section “4.5. Doping with metals the role in enhancement the photocatalytic activity of TiO2 (lines 663-685).

8-The author should talk about doping TiO2 with rare earth elements.

Answer

Thanks a lot for your comment. Was made the new section “4.4.3 Doping TiO2 with rare earth elements” (lines 639-661)

4.4.3 Doping TiO2 with rare earth elements

In recent efforts to enhance the photocatalytic properties of titanium dioxide (TiO2), the strategy of doping with rare earth elements has emerged as a promising avenue. Rare earth elements, including cerium (Ce)[137], lanthanum (La)[138], and neodymium (Nd)[139], exhibit unique electronic configurations that can influence the electronic structure and surface properties of TiO2 [140]. Doping TiO2 with rare earth elements has been shown to enhance light absorption in the visible region, thereby extending its photoresponse range. The introduction of rare earth elements can also improve charge carrier separation and migration, mitigating the recombination of electron-hole pairs, which is a common limitation in traditional TiO2 photocatalysis [141].

Several studies have reported the positive impact of rare earth element doping on TiO2 photocatalysis. For example, Ce-doped TiO2 nanomaterials have demonstrated improved photocatalytic activity in the degradation of organic pollutants under visible light irradiation[142] . Similarly, La-doped TiO2 photocatalysts have exhibited enhanced efficiency in the reduction of carbon dioxide (CO2) to valuable hydrocarbons [143]. The incorporation of rare earth elements into TiO2 introduces unique electronic states and defect structures, influencing the material's surface chemistry and catalytic performance [16].

In conclusion, the doping of TiO2 with rare earth elements represents an evolving and promising strategy to tailor the photocatalytic properties of TiO2 for improved performance in various applications.

9-Updated the list of references.

Answer

I appreciate your suggestion. The references were updated (see the main document).

Round 2

Reviewer 1 Report

Comments and Suggestions for Authors

The authors have answered all my comments and the review can be accepted in its present form.

Comments on the Quality of English Language

Minor editing of English language required.